# Genetic Alterations in Patients with *NF2*-Related Schwannomatosis and Sporadic Vestibular Schwannomas

**DOI:** 10.3390/cancers17030393

**Published:** 2025-01-24

**Authors:** Jules P. J. Douwes, Ronald van Eijk, Sybren L. N. Maas, Jeroen C. Jansen, Emmelien Aten, Erik F. Hensen

**Affiliations:** 1Department of Otorhinolaryngology—Head and Neck Surgery, Leiden University Medical Center, 2333 ZA Leiden, The Netherlands; j.c.jansen@lumc.nl (J.C.J.); e.f.hensen@lumc.nl (E.F.H.); 2Department of Pathology, Leiden University Medical Center, 2333 ZA Leiden, The Netherlands; r.van_eijk@lumc.nl (R.v.E.); s.l.n.maas@lumc.nl (S.L.N.M.); 3Department of Pathology, Erasmus MC Cancer Institute, University Medical Center Rotterdam, 3015 CN Rotterdam, The Netherlands; 4Department of Clinical Genetics, Leiden University Medical Center, 2333 ZA Leiden, The Netherlands; e.aten@lumc.nl

**Keywords:** acoustic neuroma, cerebellopontine angle tumor, clinical genetics, *NF2*-related schwannomatosis, schwannomatosis, skull base surgery, tumor genetics, vestibular schwannoma

## Abstract

Vestibular schwannomas are tumors that occur on the nerve responsible for hearing and balance, the vestibulocochlear nerve. These benign tumors may appear on one side (unilateral) or both sides (bilateral), with bilateral cases often linked to a condition called *NF2*-related schwannomatosis. This study explored the genetic causes of vestibular schwannomas in three groups: patients with NF2, younger patients with a unilateral tumor, and older patients with a unilateral tumor. By analyzing DNA from the tumors and blood, we aimed to understand the genetic mutations that drive tumor development and how the genetics differed between the three groups. In all patients, it was found that one gene, *NF2*, played a major role in the development of vestibular schwannomas. However, there were also differences found in the types of genetic mutations and mechanisms involved, offering new insights into how these tumors form and progress.

## 1. Introduction

Vestibular schwannomas (VSs) are benign nerve sheath tumors affecting the vestibular branch of the eighth (vestibulocochlear) cranial nerve. Globally, the annual incidence of vestibular schwannomas has been estimated at 1 in 64,000 to 92,000 people, resulting in a lifetime risk of 1 in every 1000 individuals [1,2,3]. Common symptoms that vestibular schwannomas present with are hearing loss, tinnitus, and vestibular complaints [4,5]. These may be impacted by determinants such as tumor size, intrameatal and/or extrameatal tumor location, cystic elements, and involvement of the cochlea and brainstem [6]. Management options include active surveillance, surgical resection, and radiotherapy [7]. Therapeutic interventions achieve overall adequate tumor control; however, vestibular schwannomas are associated with morbidity and a reduced quality of life because of the sequalae of the disease itself and/or the interventions performed [8,9,10].

Sporadic vestibular schwannomas typically arise unilaterally (uVSs). Bilateral vestibular schwannomas (bVSs), in contrast, are the hallmark lesion for *NF2*-related schwannomatosis (NF2) [11], a tumor syndrome predisposing one to multiple central nervous system tumors, including schwannomas, meningiomas, and ependymomas [11,12]. In the majority of cases, vestibular schwannoma is unilateral and solitary, but in 5% to 7%, it is associated with NF2 [1,13]. In contrast to uVS, NF2-related bVS may develop at a younger age, show a higher tumor progression rate, and respond less favorably to standard treatment options. As a result, patients with NF2 experience a higher morbidity, a higher mortality, and a more substantial decline in their quality of life [14,15,16]. The clinical severity of NF2, however, is highly variable. According to classical phenotype models, NF2 can present as a more severe (Wishart) phenotype with an early age of onset and a milder (Feiling–Gardner) phenotype with a later onset [17]. More detailed genotype–phenotype correlations have been proposed to predict the clinical severity of NF [16,18,19]. Other molecular measures, such as dysregulated cellular pathways and tumor microenvironment, are also coming to the forefront to provide greater prognostic insight [20,21,22].

The tumorigenesis of both uVS and bVS is hypothesized to be the result of a two-hit mutational mechanism, most frequently caused by inactivation of the neurofibromatosis type 2 (*NF2*) tumor suppressor gene, either by mutation of the *NF2* or loss of chromosome 22q (LOH) [23,24,25,26]. In the majority of tumors, a pathogenic nucleotide variant in *NF2* is detected together with the loss of the *NF2* wildtype copy of chromosome 22q. In *NF2*-related schwannomatosis, constitutional frameshift and nonsense (truncating) variants are linked to an aggressive phenotype originating at an adolescent age. Missense, in-frame, and silent variants have been associated with a milder phenotype and greater life expectancy, while splice-site and (multiple) exon-deleting variants result in a more variable clinical severity [16,18,27]. Vestibular schwannomas have also been associated with variants in several other genes, such as *CDC27*, *CDKN2A*, *LZTR1*, *MDM2*, *PRKAR1A*, *SMARCB1*, *TSC1,* and *TSC2* [20,24,26,27,28,29,30,31,32,33,34,35].

Although they are distinct disease types, there is both a clinical and genetic overlap between patients with uVS and patients with NF2-related bVS. It seems logical to suggest that young patients with NF2 develop uVS before they develop bVS, while patients under 30 years old presenting with uVS have a greater risk of developing bVS at a later point in life [36,37]. Testing for NF2 is therefore imperative in young patients less than 30 years of age with schwannomas of any location. It has also been suggested that uVS can be a milder phenotypical expression of NF2, as mosaicism rates of de novo NF2 have been found to be 60%, substantially higher than previously expected [19,38]. At the same time, true uVS typically develops after the age of 40 years [1,39]. Therefore, differences in the phenotype of patients with NF2 and patients with uVS, but also within-group differences for uVS, may be the result of distinct mutational mechanisms [40].

The aim of this study was to evaluate the causative genetic alterations of vestibular schwannomas in different patient groups, i.e., patients with confirmed NF2, young patients with uVS (≤30 years), and older patients with uVS (≥40 years).

## 2. Materials and Methods

### 2.1. Participants and Study Design

The present study included patients that were surgically treated for a vestibular schwannoma at the Leiden University Medical Center (LUMC) in The Netherlands, a tertiary skull base and NF2 referral center, between 1 December 1997 and 31 December 2023. The inclusion criteria comprised patients ≥ 18 years old with NF2-related or unilateral vestibular schwannoma and for whom frozen or fixated paraffine tumor material was available for retrospective genetic analysis. The diagnosis of NF2 was confirmed using the updated clinical and radiologic criteria [12]. The exclusion criteria comprised insufficient demographic or clinical data, tumor material with a lack of qualitative or quantitative DNA for genetic analysis, and other schwannomatosis-related disease types.

Following ethical approval and informed consent, electronic patient files were retrospectively examined for demographic, clinical, and genetic characteristics. Tumor specimens for variant analysis were collected from the Department of Pathology (LUMC), stored for long-term safekeeping in accordance with national guidelines. A secured online database was set up for the collection and storage of data. The study protocol was conducted in compliance with the ethical standard of The Declaration of Helsinki (2013).

### 2.2. Demographics and Clinical Assessment

The demographics included vital status, sex, age at the time of diagnosis, and age at the time of surgical intervention. For the present analysis, patients were divided into three groups: patients with clinically or genetically confirmed diagnosis of *NF2*-related schwannomatosis, patients ≤ 30 years old when diagnosed with uVS, and patients ≥ 40 years old when diagnosed with uVS. Clinical outcomes of interest included tumor characteristics, treatment history, family history of NF2 or uVS, and genetic germline diagnostics (in the case of NF2 or individuals at risk of NF2 [41]).

### 2.3. DNA Extraction and Analysis

Germline DNA was extracted from peripheral blood lymphocytes, and vestibular schwannoma DNA was extracted from micro-dissected frozen or paraffine tumor specimens, using standard diagnostic procedures. In some cases, a genetic analysis had already been performed on lymphocyte DNA and/or tumor specimens prior to this study at the discretion of the treating multidisciplinary team or in compliance with patient preferences [42].

Since 2017, next-generation sequencing (NGS) has been used as the golden standard for detecting pathogenic variants in clinical settings [43]. Prior to that, direct Sanger sequencing and multiple ligation-dependent probe amplification (MLPA) were employed. Although NGS is favored over Sanger sequencing, we included all individuals that had received germline and/or tumor analysis to deal with sample size constraints. All the included samples were sequenced and analyzed according to a globally recognized standard for quality management and handled according to medical ethical guidelines described in the Code Proper Secondary Use of Human Tissue established by the Dutch Federation of Medical Sciences [44].

The LUMC Department of Pathology (section Molecular Diagnostics) designed an Ion-Torrent-based NGS Ampliseq gene panel (Thermo Scientific, Waltham, MA, USA), named Rare Cancer PaneL (RCPL) [45]. The RCPL sequences genes and hotspot regions in rare tumors that frequently express distinct mutational profiles. For vestibular schwannomas, the RCPL includes, among others, the following relevant genes: *CDC27*, *CDKN2A*, *LZTR1*, *MDM2/CDK4*, *NF2*, *PRKAR1A*, *SMARCB1*, *SMARCE1*, *TSC1*, and *TSC2*. Additionally, LOH were assessed for chromosome 22. The sequencing criteria have been described in more detail in the RCPL guideline [45].

Outcomes of interest included the affected gene, pathogenicity classification according to the American College of Medical Genetics and Genomic [46], variant type, variant effect, location of the variants according to predetermined *NF2* gene domains (exon 1, exon 2–7, exon 8–13, and exon 14–17) [16,18], and presence of LOH. If applicable, a genetic severity score was established [18].

### 2.4. Statistics

For the statistical analyses, IBM SPSS Statistics for Windows (version 27.0. Armond, NY, USA: IBM Corp) was used. Descriptive statistics were used to describe patient characteristics. Non-normally distributed data were reported using median, range, frequency count, and percentages. Missing, unused, and incorrect data were documented and adjusted for during data collection and study analysis.

## 3. Results

### 3.1. Patient Characteristics

Ninety-three patients were included. Of these, 16 patients (17%) were diagnosed with NF2, 36 patients (39%) were ≤30 years old with uVS, and 41 patients (44%) were ≥40 years old with uVS (Table 1). The median age at diagnosis was 33.7 years (12.6–61.9) for NF2, 27.1 (16.7–30.6) for young patients with uVS, and 55.5 (40.0–79.5) for older patients with uVS. Fifteen patients with NF2 (94%) presented with bilateral vestibular schwannomas. Three patients with NF2 (19%) passed away after being diagnosed, and no deaths were recorded among the patients with uVS.

### 3.2. Germline Analysis

Germline sequencing of *NF2* was performed in twelve patients with NF2 (75%) prior to this study. Variant analysis was primarily performed with a combination of Sanger sequencing and MLPA (8/12, 67%), followed by Sanger sequencing alone (2/12, 17%) and NGS (1/12, 8%). For one case, the sequencing technique was unknown.

In five patients with NF2 (42%), a pathogenic germline variant was detected. Their median age at presentation was 24.3 years (12.5–30.6). Patient 7 had a large deletion of 19 genes on chromosome 22, including exon 1 of the *NF2* gene. No pathogenic variants were detected in *LZTR1* and *SMARCB1*. The other four patients had different nucleotide variants of *NF2*, all classified as likely pathogenic (Human Genome Variation Society Nomenclature class 4) [47]. The molecular details are described in Table 2.

Twenty-three young patients with uVS (66%) underwent germline NF2 screening [36]. We deployed Sanger sequencing with MLPA in fifteen cases (65%), NGS in seven cases (30%), and a sequencing technique of unknown origin in a single case (4%). All patients (100%) were tested for *NF2*, while fourteen patients (61%) were additionally tested for *LZTR1* and five (22%) for *SMARCB1* alterations. One patient (4%) harbored an *LZTR1* variant (NM_006767.3 (exon 13): c.1433G > A, p.Arg478Gln), yet pathogenicity was unknown (class 3, variant of unknow significance (VUS)). No other variants were detected.

### 3.3. Tumor Analysis

Tumor samples were primarily analyzed using NGS (NF2: 94%; uVS ≤ 30 years: 92%; and uVS ≥ 40 years: 100%), specifically with the RCPL (NF2: 89%; uVS ≤ 30 years: 94%; and uVS ≥ 40 years: 100%). The remaining samples were sequenced using Sanger sequencing and MLPA.

In all patients with NF2 (100%), two or more hits were detected in the tumor DNA. The detection rate was marginally lower for patients with uVS: at least 1 hit in 32 out of 36 tumors (89%) among young patients with uVS and 40 out of 41 tumors (98%) in older patients with uVS. Patients with *NF2*-related schwannomatosis harbored an even distribution of tumor samples with either two distinct nucleotide variants or one variant as the presumed first hit and the LOH of chromosome 22 as the second hit. The majority of patients with uVS possessed the latter (Figure 1).

Two patients with NF2 harbored three hits: a germline *NF2* variant, a second hit in *NF2,* and a third *SMARCB1* VUS or LOH of chromosome 22, respectively. Among the patients with uVS, three individuals aged ≤30 years and four aged ≥40 years showed a three- or four-hit mechanism in the tumor sample. A relatively high number of nucleotide variants were found in the NF2-related samples, whereas the samples from patients with uVS showed more LOH events (Figure 2). The screening details are available online in the Appendix A.

Nucleotide variants were primarily identified in *NF2* in comparison to other genes (Table 3). Tumors from younger patients with uVS harbored the greatest diversity of variants, namely in *NF2* (*n* = 40; 85%), *SMARCB1* (*n* = 3; 6%), *TSC1* (*n* = 2; 4%), *LZTR1* (*n* = 1; 2%), and *NF1* (*n* = 1; 2%). In the *NF2* gene, a substitution was predominant in patients with NF2 compared to a frameshift deletion in older patients with uVS. Nonsense variants were most frequently identified in NF2-related tumors, while frameshift variants predominated in patients with uVS.

All five patients with NF2 that had an identifiable germline variant also had a second NF2-related pathogenic hit in the tumor (Table 4). In two out of the five tumors (40%), a truncation was identified.

### 3.4. Age Differences in NF2

An additional subgroup analysis was performed on the NF2 cohort, based on their age of diagnosis in relation to the group’s median age. Young patients with NF2 (<33.7 years) had a more severe clinical phenotype compared to older patients with NF2 (>33.7 years) with respect to their vital status (deceased, % = 38 vs. 0), the presence of meningiomas (patients, % = 100 vs. 38), and the total tumor load (tumor count, median = 15 vs. 5).

The genetic analysis showed that younger patients with NF2 were more likely to harbor a nucleotide variant in *NF2* as the second hit (patients, % = 63 vs. 38), had a higher variant pathogenicity (classification, mean = 4.1 vs. 3.8), and carried more truncating variants in the tumor (variants, % = 73 vs. 50). On the other hand, LOH was less prevalent in the younger NF2 subgroup (patients, % = 50 vs. 71).

### 3.5. Sequencing Statistics

An average sequencing coverage of 94.6% (range 92.3–98.9%) for the coding sequences of *NF2*, *LZTR1*, and *SMARCB1* was achieved. Targeted bases were sequenced at least 150 times with an average of >1000 reads. Further details are provided in the RCPL source file [45].

## 4. Discussion

In the present study, causative genetic alterations of vestibular schwannomas were investigated from patients with NF2, young patients with uVS (age ≤ 30 years), and older patients with uVS (age ≥ 40 years). All patients showed pathogenic gene variants located in *NF2*, underlining its central role as a driver in the pathogenesis of NF2-related and sporadic vestibular schwannomas [24,27].

However, our results also show genetic differences between the study groups. Patients with *NF2*-related schwannomatosis had an even distribution of samples with either two distinct *NF2* variants or one *NF2* variant and LOH of chromosome 22 (44%), whereas patients with uVS predominantly possessed one *NF2* variant with LOH (53–59%). The frequency of LOH also varied across study groups. Previously reported rates in patients with uVS ranged between 53% and 75% [24,27,31,37,48]. In patients with NF2, a slightly lower range of 45% to 67% was described [26,27,49]. Here, we detected similar rates of LOH: 62% for NF2-related tumors, 68% for young patients with uVS, and up to 73% in older patients with uVS. These findings support the hypothesis that biallelic inactivation of the *NF2* gene is required for the tumorigenesis of vestibular schwannoma, yet they also point out possible different mutational pathways observed in patients with NF2 and patients with uVS [24,25,26,31]. One possible explanation may be the unique timing of mutational events, as patients with NF2 already harbor pathogenic *NF2* variants during the embryonic stage. However, other explanations such as intronic variants, copy-neutral LOH, or technical limitations of panel-based NGS should also be considered.

In total, nucleotide variants constituted 76% of the mutational hits in patients with NF2, 69% in younger patients with uVS, and 64% in older patients with uVS. As with most tumor-suppressor genes, most mutations in *NF2* were predicted to truncate (by nonsense or frameshift) or shorten (by incorrect splicing) the protein product. Truncating variations were most prevalent across all three study groups, yet a difference was observed in the truncation type. Nonsense variants were predominant over frameshift variants in patients with NF2, in line with previous studies on patients with NF2 [19,50]. On the other hand, frameshift variants were more prevalent in patients with uVS and the ratio of frameshift to nonsense variants substantially increased with an older age in patients with uVS [26,37,48]. This is in agreement with the hypothesis that age-related increase in mutagenesis and decreased potential for DNA repair lead to an altered mutational pattern [37,41,48]. Splice-site *NF2* variants are the second most common mutational hit after truncation. This was also visible in our data. Interestingly, substantial phenotypic variability can be observed between splice-site variants [16,18,51,52]. Our sample size was too small to test for such phenotypic differences. Missense variants are less common, but occur somewhat more frequently in NF2-related tumors than their unilateral counterpart [53,54].

The exon location of pathogenic variants alone has also been linked to specific NF2 genotype–phenotypes, with variants detected towards the 5′ end of *NF2* being associated with a more severe phenotype [18]. In uVS, a similar trend towards the 5′ end has been observed [55]. In all three our study groups, the majority of patients harbored nucleotide variants of *NF2* towards the 5′ end. A correlation with their clinical phenotype was beyond the scope of this research.

In a minority of tumors found across all patient subgroups, the pathogenic gene variants in *NF2* were accompanied by variants in other genes, such as *LZTR1* and *SMARCB1,* two candidate genes for an alternative diagnosis of 22q-related schwannomatosis (SWN), and tumor suppressor gene *TSC1*, which may function as a co-factor in tumorigenesis. While this highlights the genetic heterogeneity in the tumorigenesis of vestibular schwannomas [33,35], the detection rates of genetic variants in genes other than *NF2* remain low and vary across studies [33,35,48,50,56].

This study has several limitations. All patients in the present study received surgery, thus introducing selection bias towards a more severe phenotype which required surgical intervention. Additionally, the sample size of the groups of patients with NF2 was relatively small, inherent to the rarity of NF2. We performed statistical comparisons for all variables possible, yet, due to a lack of power, we were not able to demonstrate statistically significant results. Furthermore, the reliance on archival tumor samples (the oldest sample dating back to 1997) for genetic analysis may have introduced potential biases related to sample quality, tissue preservation methods, and variations in sequencing techniques over time. Lastly, variability in sequencing techniques across patients could have introduced underestimations or inconsistencies in variant detection and interpretation.

Despite these limitations, several strengths can also be identified. First, the inclusion of three distinct patient groups—patients with NF2, younger patients with uVS (≤30 years), and older patients with uVS (≥40 years)—provided a valuable evaluation of vestibular schwannomas across different age groups and disease subtypes. Second, the incorporation of both germline and tumor DNA analysis provided insights into both hereditary and somatic genetic events contributing to vestibular schwannoma tumorigenesis.

*NF2*-related schwannomatosis is a disease characterized by considerable clinical variation, with patients who present with progressive tumors at a young age and others who present with limited, slowly progressive disease at an older age. This is highlighted by the genetic differences found in the present young and old NF2 subgroups. In the present study, no young patient with uVS harbored a germline *NF2* variant. The mutational hits and number of mutational events appeared to be more similar to older patients with uVS than to patients with NF2. Therefore, the expectation is that few young patients with uVS actually harbored *NF2*-related schwannomatosis. A follow-up analysis with age-matched patients with uVS could provide further insights into the unique genetic characteristics of vestibular schwannomas across disease subtypes and severities, while it may also help predict the risk of developing bVS at a later point in life.

## 5. Conclusions

Our study contributes to a better understanding of the genetic alterations in vestibular schwannomas across different disease subtypes and age groups. In all three study groups, biallelic inactivation of the *NF2* gene was the main cause of vestibular schwannoma tumorigenesis. However, in patients with NF2, we found an even distribution of either pathogenic gene variants in both *NF2* copies or one pathogenic variant together with the LOH of chromosome 22. Meanwhile, in patients with uVS, we most frequently observed a single pathogenic *NF2* variant in combination with LOH. In addition, truncating variations were the most prevalent across subgroups, yet a difference was observed in the truncation type: nonsense variants were over-represented in patients with NF2, whereas frameshift variants were more prevalent in uVS.

## Figures and Tables

**Figure 1 cancers-17-00393-f001:**
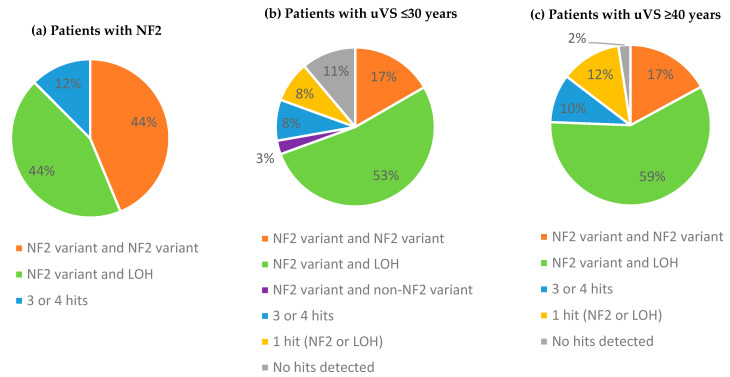
Multi-hit mechanism for the development of vestibular schwannoma in (**a**) patients with NF2 (*n* = 16), (**b**) young patients with uVS (≤30 years) (*n* = 36), and (**c**) older patients with uVS (≥40 years) (*n* = 41). Legend: patients with NF2 = patients with *NF2*-related schwannomatosis; uVS = unilateral vestibular schwannoma; and LOH = loss of heterozygosity.

**Figure 2 cancers-17-00393-f002:**
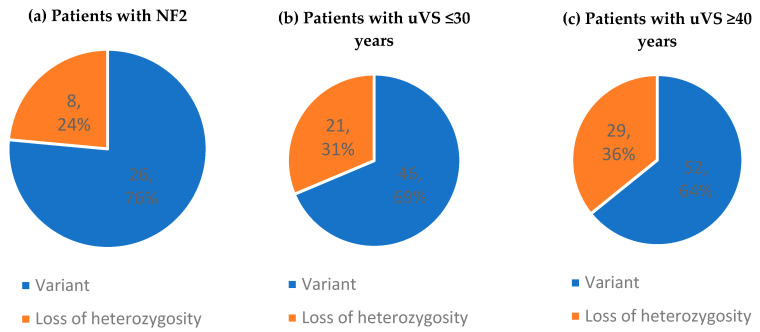
Total number of mutational events detected in vestibular schwannomas in (**a**) patients with NF2, (**b**) young patients with uVS (≤30 years), and (**c**) older patients with uVS (≥40 years). Legend: NF2 = *NF2*-related schwannomatosis; and uVS = unilateral vestibular schwannoma.

**Table 1 cancers-17-00393-t001:** Demographics and treatment characteristics of the study cohort.

Patient Characteristics	Patients with NF2, *n* = 16	Patients with uVS ≤ 30 Years, *n* = 36	Patients with uVS ≥ 40 Years, *n* = 41
Female, *n* (%)	11 (68.8)	24 (66.7)	20 (48.8)
Age, years, median (range)			
Diagnosis	33.7 (12.6–61.9)	27.1 (16.7–30.6)	55.5 (40.0–79.5)
Time of treatment	35.0 (20.7–62.7)	28.0 (17.0–38.4)	56.4 (43.7–81.0)
Current	54 (25–79)	32 (18–44)	58 (45–84)
Family history of NF2 or uVS, *n* (%)	1 (6.7)	0 (0.0)	1 (2.4)
Previous treatment on target tumor, *n* (%)			
Surgical resection	2 (12.5)	2 (5.6)	0 (0.0)
Radiotherapy	0 (0.0	5 (13.9)	0 (0.0)
Bevacizumab	2 (12.5)	n.a.	n.a.
Surgical indication, *n* (%)			
Large tumor size	10 (71.4)	22 (61.1)	11 (26.8)
Tumor progression	2 (14.3)	9 (25.0)	23 (56.1)
Symptoms	2 (14.3)	5 (13.9)	7 (17.1)

Legend: NF2 = *NF2*-related schwannomatosis; *n* = number of patients; uVS = unilateral vestibular schwannoma; and n.a. = not applicable.

**Table 2 cancers-17-00393-t002:** Pathogenic germline variants in five patients with *NF2*-related schwannomatosis.

No.	Age at Pres.	Exon	DNA Sequence	Protein Sequence	Variant Type	Genetic Severity
02	24.3	12	c.1132G > T	p.Glu378*	Nonsense	2B
04	16.5	8	c.702dupT	p.Gly235Trpfs*	Frameshift	3
06	12.6	13	c.1341-2A > G	p.?	Splice site	2A
07	30.6	1	c.del22q12.1q12.2	p.?	Large deletion	2A
15	27.2	10	c.892C > T	p.Gln298*	Nonsense	2B

Legend: No. = patient study number; and Pres. = presentation.

**Table 3 cancers-17-00393-t003:** Detailed description of nucleotide variants identified in vestibular schwannomas.

Variants	Patients with NF2	Patients with uVS ≤30 Years	Patients with uVS ≥40 Years
Gene, *n* (%)			
NF2	25 (96.2)	40 (85.1)	48 (92.3)
LZTR1	0	1 (2.1)	0
SMARCB1	1 (3.8)	3 (6.4)	2 (3.8)
Other	0	3 (6.4)	2 (3.8)
Type of *NF2* variant, *n* (%)			
Substitution	14 (56.0)	17 (48.6)	18 (38.3)
Duplication	1 (4.0)	2 (5.7)	3 (6.4)
Insertion	0	0	1 (2.1)
Deletion	8 (32.0)	14 (40.0)	23 (48.9)
Deletion/insertion	1 (4.0)	2 (5.7)	2 (4.3)
Large deletion	1 (4.0)	0	0
Unknown (c.?)	0	4	1
Effect of *NF2* variant, *n* (%)			
Missense	2 (8.3)	1 (2.9)	2 (4.4)
Splice site	5 (20.8)	4 (11.8)	6 (13.3)
Frameshift	7 (29.2)	15 (44.1)	23 (51.1)
Nonsense	9 (37.5)	13 (38.2)	12 (26.7)
Deletion	1 (4.2)	0	0
Deletion/insertion	0	1 (2.9)	1 (2.2)
Silent	0	0	1 (2.2)
Unknown (p.?)	0	5	3
Variant location on *NF2*, *n* (%)			
Exon 1	1 (4.0)	2 (5.1)	4 (8.3)
Exons 2–7	12 (48.0)	16 (41.0)	22 (45.8)
Exons 8–13	11 (44.0)	15 (38.5)	15 (31.3)
Exons 14–16	1 (4.0)	6 (15.4)	7 (14.6)

Pathogenicity classification according to the American College of Medical Genetics and Genomics [47]. Legend: NF2 = *NF2*-related schwannomatosis; uVS = unilateral vestibular schwannoma; and *n* = number of cases.

**Table 4 cancers-17-00393-t004:** Pathogenic tumor variants in patients with *NF2*-related schwannomatosis with identified germline NF2 variants.

No.	First (Germline) *NF2* Hit ^1^	Second Hit in Tumor	Third Hit in Tumor
02	c.1132G>, p.Glu378*	NF2 (exon 12): c.558_561delGAGA, p.Arg187Leufs*	LOH
04	c.702dupT, p.Gly235Trpfs*	*NF2* (exon 12): c.1331C > A, p.Ser444*	n.a.
06	c.1341-2A > G, p.?	LOH	n.a.
07	c.del22q12.1q12.2, p.?	*NF2* (exon 7): c.632C > A, p.Ala211Asp	n.a.
15	c.892C > T, p.Gln298*	LOH	n.a.

^1^ Germline details in Table 2. Legend: No. = patient number; LOH = loss of heterozygosity; and n.a. = not applicable.

## Data Availability

The original contributions presented in this study are included in the Appendix A. Further inquiries can be directed to the corresponding authors.

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
