# Peer review of "Genetic Alterations in Patients with NF2-Related Schwannomatosis and Sporadic Vestibular Schwannomas"

_cancers, 2025, doi:10.3390/cancers17030393_

Round 1
Reviewer 1 Report
Comments and Suggestions for Authors
Congratulations! This is a sound scientific paper. I wish you had specualted a little about clinical relevance and possible genetic targets for therapeutical intervention, but since you refrained from doing so, you made this manuscript unassailable ;-)
Author Response
“Congratulations! This is a sound scientific paper. I wish you had speculated a little about clinical relevance and possible genetic targets for therapeutical intervention, but since you refrained from doing so, you made this manuscript unassailable.”
It would indeed be of great interest to link the genetic findings of the study to clinical outcomes or possible therapeutic interventions, yet we believe this would be somewhat speculative given our current study design. That said, we thank the reviewer for their positive feedback.
Reviewer 2 Report
Comments and Suggestions for Authors
Dear authors congratulations on the manuscript.
Below are some pointers for further improvement so that it can be reconsidered for publication.
1)A load-bearing limitation is definitely the time frame, in 1985 years of the beginning of the series series we did not have the same molecular knowledge as today we need to better clarify therefore the inclusion criteria.
2)the introduction could consider the classification of VS.
3) the article is too unbalanced on the molecular aspect, it would be useful and also offer guidance to the scientific community about surgical approaches and morbidities, VS should be compared with other shwannomas, especially TS in which association with NF mutation persists. In this regard, I point you to read and cite two articles: https://doi.org/10.3390/jcm13133701 ; https://doi.org/10.1007/s00701-024-06292-8
4) Introduce a paragraph with limitations before conclusions.
I look forward to reading the revised manuscript, please be sure to follow my instructions.
Comments on the Quality of English Language
minor editing needed
Author Response
“Dear authors, congratulations on the manuscript.
Below are some pointers for further improvement so that it can be reconsidered for publication. I look forward to reading the revised manuscript, please be sure to follow my instructions.
Point 1 - A load-bearing limitation is definitely the time frame, in 1985 years of the beginning of the series we did not have the same molecular knowledge as today. We need to better clarify therefore the inclusion criteria.
We thank the reviewer for highlighting this important point. We agree that the timeframe is a significant limitation of the study, and we had already addressed this as a limitation in the manuscript (lines 300–302). However, although the first patient in our cohort was included in 1985, this patient was not operated on until 2008. The oldest tumor sample in our study actually dates back to December 1997, reducing the timeframe by 12 years. We have clarified this in the Methods section (line 98) and added a brief statement regarding the age of the oldest sample (line 300-301).
Additionally, we have added two statements to clarify extensive measures were taken to ensure that the quality of the specimens used in the study was suited for genetic analysis, despite some samples being of an older age (lines 131-132, lines 141-142).
Point 2 - The introduction could consider the classification of VS.
In response to the reviewer’s suggestion, we have added the following passage to the Introduction: “These may be impacted by determinants such as tumor size, intrameatal and/or extra-meatal tumor location, cystic elements, and involvement of the cochlea and brainstem.” (lines 47-48).
Point 3 - The article is too unbalanced on the molecular aspect, it would be useful and also offer guidance to the scientific community about surgical approaches and morbidities. VS should be compared with other schwannomas, especially TS in which association with NF mutation persists. In this regard, I point you to read and cite two articles: https://doi.org/10.3390/jcm13133701; https://doi.org/10.1007/s00701-024-06292-8
We thank the reviewer for the comment and for highlighting the link between molecular and clinical considerations in the management of vestibular schwannoma.
Available literature and research on (NF2-related) schwannomatosis tends to focus on clinical outcomes and therapeutic interventions. However, diagnostic and prognostic approaches, such as tumor (micro)biology or genetic analyses, are equally important to fully comprehend the development, diagnosis and management of schwannomatosis. Therefore, we believe one of the strengths of our study lies in the clear molecular focus.
The inclusion of non-vestibular schwannomas or a genotype-phenotype correlation falls outside the scope of the present study. Still, it would be of value to conduct a follow-up study to determine if there is a link between NF2 mutation patterns of vestibular schwannomas and non-vestibular schwannomas. Current available literature suggests that there is a stronger correlation between the genetic severity of NF2 and vestibular schwannomas compared to trigeminal schwannomas (doi: https://doi.org/10.1038/s41431-021-01029-y).
Point 4 - Introduce a paragraph with limitations before conclusions.
We would like to refer the reviewer to lines 295-304, where we have stated the limitations of the study. In the current version, this paragraph has been revised in line with comments of the reviewers:
This study has several limitations. All patients in the present study received surgery, thus introducing selection bias towards a more severe phenotype that required surgical intervention. Additionally, the sample size of the NF2 patient groups is relatively small, inherent to the rarity of NF2. We performed statistical comparisons for all variables possible, yet due to a lack of power we were not able to demonstrate statistically significant results. Furthermore, the reliance on archival tumor samples (the oldest sample dating to 1997) for genetic analysis may introduce potential biases related to sample quality, tissue preservation methods, and variations in sequencing techniques over time. Lastly, variability in sequencing techniques across patients could introduce underestimations or inconsistencies in variant detection and interpretation.
Reviewer 3 Report
Comments and Suggestions for Authors
The present study addresses, according to the aims, important issues in the management of vestibular schwannomas. Specifically the recognition of different phenotypes genotypes of NF2RLS giving rise to different phenotypes eg different morbidity is of interest for clinical management. Obviously In the present study the numbers are too few to address this question . Also the possibility of NF 2 related schwannomatosis in young patients with unilateral vestibular schwannomas is clinically pertinent. Despite not being able to add any further explanation to the latter due to the limited number o samples the results indicate different mechanisms of inactivation of the NF2 gene in NF2RLS and sporadic VS. However these assumptions are not supported by any statistical comparisons other than descriptive data.
Major issues
1. Although the numbers are small a statistical comparison between the frequencies of gene alterations would tell the reader whether the differences are significant or not. Otherwise one would speculate whether there are any significant differences between the groups other than between NF2RS and unilateral VS.
2. Is the assumption that a part of the young patients with uVS would actually be NFRS supported by the results, this is not directly addressed in the discussion?
Minor issues
1. In the discussion of clinical severity and growth of VS the horizons could be widened and discus alternative or complimentary mechanisms as, see for example< https://doi.org/10.1038/s41467-023-37226-0, https://doi.org/10.1038/s41467-023-42762-w.
2. On page2, line 76 it is stated . there is an overlap between uVS and NF2- 76 related bVS. It is not altogether clear in what respect the entities overlap.
3. Page 4 line 166-167. There seems to be some words missing in the sentence; No pathogenic variants were detected LZTR1 and SMARCB1
Author Response
“The present study addresses, according to the aims, important issues in the management of vestibular schwannomas. Specifically the recognition of different phenotypes genotypes of NF2RLS giving rise to different phenotypes eg. different morbidity is of interest for clinical management. Obviously in the present study the numbers are too few to address this question. Also the possibility of NF 2 related schwannomatosis in young patients with unilateral vestibular schwannomas is clinically pertinent. Despite not being able to add any further explanation to the latter due to the limited number of samples the results indicate different mechanisms of inactivation of the NF2 gene in NF2RLS and sporadic VS. However these assumptions are not supported by any statistical comparisons other than descriptive data.
Major issues
Point 1 - Although the numbers are small a statistical comparison between the frequencies of gene alterations would tell the reader whether the differences are significant or not. Otherwise one would speculate whether there are any significant differences between the groups other than between NF2RS and unilateral VS.
The reviewer addresses a good point here. In fact, we did perform statistical comparisons for variables/frequencies where possible. Unfortunately, the statistical tests did not show clear correlations, even when trying to control for the small(er) numbers. Small patient numbers due to the rarity of a disease such as NF2-related schwannomatosis are a well-known challenge in these type of evaluations. The difficulty of acquiring larger NF2 patient cohorts is illustrated by the systematic review on targeted treatment for VS (https://doi.org/10.1093/noajnl/vdad099), which could only identify 10 patient cohorts totaling 200 patients, with a median number of included patients per cohort of 16.5. This is similar to our present NF2 patient group.
We discussed the matter internally and came to the conclusion that, even though the tests came back non-significant, the trends visible were still of importance as research on NF2 is limited. To prevent confusion about the possible meaning of the results we did not report non-significant statistics throughout the manuscript. Nonetheless, we agree this may still be of importance to the reader. Therefore, we added this to the study limitations (lines 298-300).
Point 2 - Is the assumption that a part of the young patients with uVS would actually be NF2RS supported by the results, this is not directly addressed in the discussion?
All 23 young uVS patients harbored a sporadic vestibular schwannoma, a major criterium for NF2-related schwannomatosis (Table 1, https://doi.org/10.1016/j.gim.2022.05.007https://doi.org/10.1016/j.gim.2022.05.007). The large majority of tumor samples (89%) also harbored at least one pathogenic variant associated with NF2-related schwannomatosis. If later in life these patients develop bVS or an identical NF2 pathogenic variant is detected in another NF2-related tumor, they do classify as NF2-related schwannomatosis.
However, as patients under 30 years old presenting with uVS have a greater risk to develop bVS at a later point in life, they all underwent genetic germline testing for NF2. None of the patients harbored a germline NF2 variant. We additionally found the mutational hits (Figure 1) and number of mutational events (Figure 2) of young uVS patients to be more similar to older uVS patients than to NF2 patients. Therefore, weighing our results and the abovementioned, the expectation is that few of the young uVS patients actually harbor NF2-related schwannomatosis.
We have elaborated on this in the last paragraph of the Discussion to address this assumption (lines 314-318).
Minor issues
Point 3 - In the discussion of clinical severity and growth of VS the horizons could be widened and discus alternative or complimentary mechanisms, see for example https://doi.org/10.1038/s41467-023-37226-0 / https://doi.org/10.1038/s41467-023-42762-w.
We thank the reviewer for these interesting suggestions and agree that the field of diagnostics and prognostics for vestibular schwannoma should not be limited to clinical or genetic outcome measures. To address this, we have added a statement and related references in the Introduction to broaden the reader's perspective (lines 65-67). However, as our study maintains a clear genotype-phenotype focus, we believe it is most appropriate to keep the aforementioned statement concise to avoid deviating from the primary scope of the manuscript.
Point 4 - On page 2, line 76 it is stated… there is an overlap between uVS and NF2-related bVS. It is not altogether clear in what respect the entities overlap.
We clarified that uVS and NF2-related bVS overlap both in clinical presentation and genetic make-up. Lines 80-81 were adjusted to make this more clear.
Point 5 - Page 4 line 166-167. There seems to be some words missing in the sentence; No pathogenic variants were detected LZTR1 and SMARCB1.
Correct, the word “in..” has been added (line 173).
Round 2
Reviewer 2 Report
Comments and Suggestions for Authors
The authors did not follow my directions, so I defer the decision to the publisher.